# T3 Critically Affects the Mhrt/Brg1 Axis to Regulate the Cardiac MHC Switch: Role of an Epigenetic Cross-Talk

**DOI:** 10.3390/cells9102155

**Published:** 2020-09-24

**Authors:** Francesca Forini, Giuseppina Nicolini, Claudia Kusmic, Romina D’Aurizio, Alberto Mercatanti, Giorgio Iervasi, Letizia Pitto

**Affiliations:** 1CNR Institute of Clinical Physiology, Via G.Moruzzi 1, 56124 Pisa, Italy; nicolini@ifc.cnr.it (G.N.); kusmic@ifc.cnr.it (C.K.); a.mercatanti@ifc.cnr.it (A.M.); iervasi@ifc.cnr.it (G.I.); letizia.pitto@ifc.cnr.it (L.P.); 2Institute of Informatics and Telematics (IIT), CNR, via G. Moruzzi 1, 56124 Pisa, Italy; romina.daurizio@gmail.com

**Keywords:** long non-coding Mhrt, Brg1, chromatin remodeling, myosin heavy chain switch, low T3 state, T3 replacement, cardiac remodeling

## Abstract

The LncRNA my-heart (Mhrt) and the chromatin remodeler Brg1 inhibit each other to respectively prevent or favor the maladaptive α-myosin-heavy-chain (Myh6) to β-myosin-heavy-chain (Myh7) switch, so their balance crucially guides the outcome of cardiac remodeling under stress conditions. Even though triiodothyronine (T3) has long been recognized as a critical regulator of the cardiac Myh isoform composition, its role as a modulator of the Mhrt/Brg1 axis is still unexplored. Here the effect of T3 on the Mhrt/Brg1 regulatory circuit has been analyzed in relation with chromatin remodeling and previously identified T3-dependent miRNAs. The expression levels of Mhrt, Brg1 and Myh6/Myh7 have been assessed in rat models of hyperthyroidism or acute myocardial ischemia/reperfusion (IR) treated with T3 replacement therapy. To gain mechanistic insights, in silico analyses and site-directed mutagenesis have been adopted in combination with gene reporter assays and loss or gain of function strategies in cultured cardiomyocytes. Our results indicate a pivotal role of Mhrt over-expression in the T3-dependent regulation of Myh switch. Mechanistically, T3 activates the Mhrt promoter at two putative thyroid hormone responsive elements (TRE) located in a crucial region that is necessary for both Mhrt activation and Brg1-dependent Mhrt repression. This newly identified T3 mode of action requires DNA chromatinization and is critically involved in mitigating the repressive function of the Brg1 protein on Mhrt promoter. In addition, T3 is also able to prevent the Brg1 over-expression observed in the post-IR setting through a pathway that might entail the T3-mediated up-regulation of miR-208a. Taken together, our data evidence a novel T3-responsive network of cross-talking epigenetic factors that dictates the cardiac Myh composition and could be of great translational relevance.

## 1. Introduction

Adverse hypertrophic remodeling is a common hallmark of a variety of cardiovascular ailments including ischemic diseases, hypertension, valvular dysfunctions, and inherited cardiomyopathies. Despite available interventions improving patient prognosis, sustained pathological hypertrophy is still one of the main risk factors for heart failure and cardiovascular mortality in humans [1]. A better understanding of the etiopathological background is necessary to develop more resolutive therapeutic strategies. 

At molecular level, the evolution of pathological hypertrophy is due to a maladaptive down-regulation of the adult isoforms of cardiac muscle proteins including the alpha myosin heavy chain (α-MHC) and a concomitant up-regulation of the fetal genes such as the beta myosin heavy chain (β-MHC), atrial natriuretic peptide (ANF), and brain natriuretic peptide (BNP) [2]. Recently a long non-coding RNA (lncRNA), termed Myheart (Mhrt), was demonstrated to protect the heart against adverse remodeling by inhibiting the maladaptive alpha to beta myosin heavy chain (MHC) switch [3]. Despite being located in the antisense strand between the α-MHC (Myh6) and the β-MHC (Myh7) genes, the Mhrt function does not involve RNA-RNA interference. Instead, Mhrt transcripts sequester the ATP-dependent Brahma-related gene 1 (Brg1) protein. Brg1 is the helicase subunit of the Brg1-associated-factors (BAF) chromatin remodeling complex and is responsible both for the negative regulation of Myh6 and for Myh7 activation [4,5]. As for MHC isoforms, in mammals the cardiac level of Mhrt and Brg1 are antithetically regulated both during development and in pathological conditions [3,4]. In the fetal life, Mhrt repression and Brg1 up-regulation are required to maintain the cardiac myocytes in an embryonic state characterized by Myh6 silencing and Myh7 up-regulation. Soon after birth, a dramatic surge of Mhrt expression and the concomitant suppression of Brg1signaling reverse such a MHC switch, fostering the postnatal differentiation and growth of the heart. Brg1 is reactivated by pathologic cardiac stress leading to repression of cardiac Myh6 and Mhrt transcription and to development of cardiomyopathy [3,5]. In this context, Mhrt over-expression protects the heart from hypertrophy and failure by preventing Brg1 to recognize its genome targets [3]. The reciprocal Mhrt/Brg1 inhibition represents a critical feedback circuit for maintaining cardiac function, therefore any intervention aimed at maintaining the physiological Mhrt/Brg1 cardiac homeostasis may be regarded as cardioprotective. 

Thyroid hormones (THs) are well-known activators of the gene expression program typical of the adult, differentiated cells. The available findings suggest that both during mammalian development and in pathological conditions, the time course of the cardiac Mhrt and Brg1 expression reflects the plasma levels of 3,5,3′-triiodothyronine (T3), the main biologically active TH. Total and free T3 levels remain low during the fetal period and show a significant surge at gestation term and throughout postnatal life [6], which coincides with the Mhrt/Brg1 and MHC isoform switches. Under cardiac disease conditions, a turning off of the TH signaling favors the maladaptive return to a fetal-like gene program responsible for pathological hypertrophy and heart dysfunction [7,8,9]. Restoring cardiac T3 levels under these conditions contributes to improving myocardial performance and protecting the heart against adverse hypertrophy and remodeling [7,10,11,12,13,14], which is associated with the normalization of the circulating BNP levels and of the α-MHC mRNA myocardial content [15].

While the antithetical regulation of Myh6 and Myh7 gene expression by T3 via its nuclear receptors has long been recognized, so far the role of the cardiac Mhrt/Brg1 axis in the T3-dependent modulation of the Myh6/Myh7 expression ratio has never been investigated. To bridge this gap, in the present study we have exploited both in vivo and in vitro animal models and characterized a new T3-dependent cardioprotective mechanism that controls MHC isoform composition and relies on the cross-talk between Mhrt, Brg1, miR-208a, and chromatin remodeling. 

## 2. Materials and Methods

### 2.1. Experimental Design and Animal Models

The experimental design included an in vivo exploratory investigation on established rat models of hyperthyroidism and acute myocardial ischemia and reperfusion (IR), followed by an in vitro validation study on neonatal rat cardiomyocytes (NRCM). The left ventricle samples used in the in vivo study were collected during previous works [16,17,18]. NRCM were obtained from rat pups as described below. All the animal procedures used in this investigation conformed to the recommendations in the Guide for the Care and Use of Laboratory Animals published by the US National Institutes of Health (NIH Publication No. 85–23, revised 1996) and the protocols were approved by the Animal Care Committee of the Italian Ministry of Health (Endorsement n.552/20156-PR) and by the local Animal Care Committee. Briefly, hyperthyroidism was induced by subcutaneous injection of 25 μg/kg/day thyroxine (T4) for 6 d in 2–3-month-old Wistar rats (HyperT group); euthyroid rats receiving saline injection were used as control (EuT group) [16]. At the end of the experimental procedure, the left ventricle was immediately frozen. Myocardial ischemia/reperfusion (IR) was induced in 2–3-month-old Wistar rats by 30 min ischemia followed by unrestrained reperfusion for 3 d or 14 d. Following IR after 24 h, rats showing a low T3 state were randomly allocated to receive 48 h infusion of saline (IR group) or T3 at the replacement dose of 3 μg/kg/die (IRT3 group) [17,18]. The choice of infusing T3 instead of T4 was dictated by the notion that under condition of low T3 syndrome the tissue local modulation of the activity of different isoforms of deiodinases seems to favor the conversion of T4 into rT3 (metabolically inactive form) rather than T3 (metabolically active form) [19]. At the end of the experimental procedure, the area at risk (AAR) of the left ventricle was immediately frozen. Samples were also obtained from correspondent left ventricle regions of sham-operated rats (sham group).

The mean plasmatic values of free T3 (FT3) and free T4 (FT4) were measured basally and at the end of each experimental procedure are reported in Appendix A, Appendix A.

### 2.2. Neonatal Rat Cardiomyocyte Cell Culture

NRCM were isolated through enzymatic digestion from hearts of 2 to 3-day-old Wistar rats using the Worthington Neonatal Cardiomyocyte Isolation System according to the manufacturer’s instructions. To prevent the overgrowth of contaminating fibroblasts, a 2 h pre-plating procedure was adopted prior to NRCM plating and 10 uM Bromodeoxyuridine was added to the growing medium. NRCM were grown in a humidified atmosphere of 5% CO_2_ at 37 °C in DMEM low glucose with 2 mM l-glutamine, 1% penicillin and streptomycin, and 10% fetal bovine serum (FBS) (complete medium). In the experiments involving T3 treatment (see below), FBS was substituted with the same amount of FBS deprived of TH with standard charcoal stripping procedures (TH-free medium). Briefly, FBS was absorbed overnight at 4 °C on dextran-coated activated charcoal (40 μg/mL serum), the suspension was centrifuged to precipitate the charcoal and the supernatant was filtered through 0.2 μm filters.

### 2.3. In Silico Analyses

#### 2.3.1. Identification of the Mhrt Coding Sequence in *Rattus norvegicus*

The bona fide Mhrt coding sequence in *Rattus norvegicus’s* genome has been inferred by sequence homology searching using Basic Local Alignment Search Tool (BLASTn; https://blast.ncbi.nlm.nih.gov/Blast.cgi) with the annotated mmu Mhrt NR_033497.1 (Appendix A). The identified region, showing 93% identity over 99% of mouse Mhrt length, has been used to design primers and SiRNA for the RT-qPCR analysis of Mhrt and Mhrt silencing (see below). Since Mhrt and Myh7 exons are partially overlapping (see Figure 1A), the Mhrt primers have been designed to ensure that no amplicon could be generated from the Myh7 mRNA and vice versa (Appendix A). SiRNA against rno-Mhrt has been designed according to Poliseno et al. [20] (Appendix A).

#### 2.3.2. miRNA Target Prediction

miRNA target prediction was performed by using the online miRWalk2.0 tool (http://zmf.umm.uni-heidelberg.de/apps/zmf/mirwalk2/). To find putative miRNA binding sites within the Brg1 complete sequence, previously reported T3-regulated miRNAs were used as input [17] and the search was restricted to miRNAs that were predicted to target both human and rat Brg1 with a binding score > 85% (see Appendix A).

#### 2.3.3. Identification of Putative Thyroid Hormone Response Elements

Putative thyroid hormone receptor consensus binding sites were identified through the online Promo search tool (http://alggen.lsi.upc.es/cgi-bin/promo_v3/promo/promoinit.cgi?dirDB=TF_6.4) using the a3–a4 regulatory region of the Mhrt promoter as input sequence and setting the maximum matrix dissimilarity rate at 15%.

### 2.4. RNA Extraction and RT-qPCR

Total RNA was extracted from tissue samples and NRCM with the miRNeasy mini kit reagent (Qiagen, Hilden, Germany) according to the manufacturer’s instructions. The cDNA was synthesized from 1 μg RNA using the QuantiTect Reverse Transcription Kit (Qiagen) as directed by the manufacturer.

For gene expression analyses, 10 ng of cDNA from cardiac tissue or NRCM were processed in triplicate in a Rotor-Gene Q real-time machine (Qiagen) using the Quantifast SYBR Green Mix (Qiagen). PCR conditions were as follows: 5 min of initial denaturation and then 40 cycles of 95 °C for 10 s, 58 °C for 20 s, 72 °C for 10 s. To assess product specificity, a melting curve analysis from 65 °C to 95 °C with a heating rate of 0.1 °C/s with a continuous fluorescence acquisition was constructed. Gene transcript values were normalized with those obtained from the amplification of Hprt, Hmbs, and Gapdh. The relative quantification of samples was performed by Rotor Gene Q-Series Software and expressed as mean ± standard error of the mean (SEM). The complete list of primer sequences is shown in Appendix A.

### 2.5. Mhrt Silencing and miR-Mimic Over-Expression

For the Mhrt knock-down (KD) experiments, 2.0 × 10^5^ NRCM/well were transfected in a 6-well plate via 5 h incubation in optimem (Gibco, Waltham, MA, USA) containing 80 nM rat SiRNA-Mhrt or control decoy. Lipofectamine 2000 (Invitrogen, Carlsbad, CA, USA) was used as transfectant according to the manufacturer’s recommendations. At the end of the transfection procedure, NRCM were grown for 48 h in TH-free medium in the absence and presence of 3 nM T3 and the cell pellets were harvested for gene expression analysis. Five biological replicates for each experimental group were used in this analysis. For the miR-mimic over-expression experiments, 2.0 × 10^5^ NRCM/well were transfected in a 6-well plate via 5 h incubation in optimem (Gibco) containing 100 nM human miR-208a mimic or control mimic (See Appendix A for mimic sequences). Lipofectamine 2000 (Invitrogen) was used as transfectant according to the manufacturer’s recommendations. Following the transfection procedure, the cells were grown for further 48 h in complete medium and then harvested for gene expression analysis. Four biological replicates for each experimental group were used in this analysis.

### 2.6. Plasmid Construction and Reporter Assays 

The dual firefly-and-renilla/luciferase-reporter gene assay was performed in order to evaluate the ability of T3 alone or in combination with Brg1 to bind the chromatinized and naked Mhrt promoter. The reporter vectors were generated by cloning the full length rat Mhrt promoter into the pREP4 (gift from Dr. Zhao of NHLBI Betesda, MD, USA) and pGL3-Control Vector (Promega, Madison, WI, USA) plasmids. The sequence of interest was amplified by PCR using the Phusion™ High-Fidelity DNA Polymerase (ThermoFisher, Waltham, MA, USA). After purification through gel-electrophoresis and extraction with the Qiaquick gel extraction kit (Qiagen), the promoter was cloned into the pGEM-T Easy vector (Promega) and sequenced to confirm that no errors were introduced during the procedures. Next the promoter was excised with Not1 and sub-cloned into the Not1 sites of the pREP4. To obtain the pGL3 construct, Mhrt promoter sequence was excised from pREP4 using NheI and XhoI enzymes and further sub-cloned into the NheI/XhoI sites within the polylinker region of the pGL3-Control Vector upstream of the luciferase gene. The screening of the colonies containing the correct insert, the construct extraction, and sequencing were performed as described above for the pGEM-T Easy vector. Various versions of each reporter vector were generated containing mutation at site A (Mut-A), site B (Mut-B) or at both sites (Mut-AB) of Mhrt promoter. Mutations were introduced with PCR-mutagenesis. The oligonucleotide sequences used for Mhrt amplification and site directed mutagenesis are reported in Appendix A.

For the reporter assays, 2.5 × 10^5^ cardiomyocytes/well were co-transfected in a 6-well plate via 5 h incubation with 500 ng Mhrt reporter vector and 100 ng renilla plasmid along with 500 ng Brg1 expression plasmid (gift from Dr. Romero Ferraro) or 500 ng empty plasmid. Aliquots of 1.5 μL Lipofectamine 2000 (Invitrogen)/well were used as transfectant according to the manufacturer’s recommendations. Then, the cells were treated with 3 nM T3 or with T3 vehicle for 48 h and processed with the Dual-Luciferase^®^ Reporter Assay System (Promega) according to manufacturer’s instructions. The luminescence was quantified using the GloMax-Multi detection system luminometer (Promega). Four to five biological replicates for each experimental group were used in this analysis.

### 2.7. Statistical Analysis

All variables satisfied the condition for parametric analysis. Differences between the means of two variables were evaluated by the Student *t*-test. Comparisons between more than two groups were performed using one-way ANOVA followed by post hoc Bonferroni test correction for multiple comparisons (IBM Spss statistics 20, Armonk, NY, USA). The results are expressed as mean ± SEM, and *p* ≤ 0.05 values were considered statistically significant. Linear regression analysis was performed setting statistical significance as *p* ≤ 0.05 (IBM Spss statistics 20).

## 3. Results

### 3.1. T3-Dependent Up-Regulation of Mhrt Greatly Contributes to the Cardiac Increase of the Myh6/Myh7 Expression Ratio

We first explored the ability of T3 to modulate the cardiac level of Mhrt and the association with the well known T3-dependent regulation of MHC isoforms. To this purpose, we identified a bona fide Mhrt coding sequence in *Rattus norvegicus* showing 93% identity with the primary sequence of the annotated mouse Mhrt [1] (Figure 1a, Appendix A). Nucleotide sequence data reported are available in the Third Party Annotation Section of the DDBJ/ENA/GenBank databases under the accession number TPA: BK013310. The cardiac expression levels of Mhrt were assessed under different in vivo and in vitro physiopathological conditions and correlated with the Myh6/Myh7 expression ratio. In the left ventricles of hyperthyroid rats a significant fivefold increase of Mhrt mRNA was observed with respect to euthyroid controls (Figure 1b). In addition, a highly significant positive correlation was obtained when Myh6/Myh7 expression ratio was plotted against Mhrt levels (Figure 1b). The mechanistic link between Mhrt and T3-dependent modulation of MHC isoforms was then explored through an in vitro KD strategy. Neonatal rat cardiomyocytes were transfected with Mhrt SiRNA or control decoy in the presence or absence of T3 treatment at physiological concentration (3nM) for 48h. In line with the in vivo data, T3 administration resulted in Mhrt overexpression, which correlated to the increased Myh6/Myh7 ratio (Figure 1c). Mhrt KD greatly prevented the stimulatory effect of T3 on both Mhrt and Myh6/Myh7 thus confirming the significant contribution of Mhrt overexpression as part of the T3 regulatory action on the MHC isoforms.

To evaluate the relevance of this mechanism under in vivo cardiac stress conditions, we exploited a previously described rat model of acute IR resembling the low T3 state observed in human cardiac disease [17,18]. Twenty-four hours after IR, the IR rats were subjected to 48 h infusion of either T3-replacement to restore euthyroidism or vehicle solution. As early as 3 d post IR, we observed a substantial reduction of Mhrt expression within the area at risk (AAR) with respect to the sham-operated rats (Figure 1d). Such alteration was evident even at 14 d post IR but was completely and persistently abolished by the timely T3 replacement (Figure 1d). Again, at each time point and in each experimental condition, Mhrt expression levels significantly correlated with Myh6/Myh7 ratio (Figure 1d). Overall, these findings indicate that the fluctuation of T3 levels in both physiological and pathological conditions critically dictate the Myh6/Myh7 values through the modulation of Mhrt.

Given the pivotal role of Brg1 as a suppressor of Mhrt expression in cardiac pathophysiology [3], we then tested if T3 replacement could also affect the Brg1expression in the IR model. As shown in Figure 2a, at 3 d post-surgery we measured a significant 50% up-regulation of Brg1 that tended to revert at 14 d. T3 replacement was paralleled by a complete normalization of the Brg1 expression levels that remained lower when compared to the IR rats even in the long-term (Figure 2a). On the contrary, T3 was unable to modulate Brg1 expression in the hyperthyroid rat group and in NRCM (Figure 2b). Based on this finding we hypothesized that an indirect T3-dependent mechanism could be at work to regulate Brg1 levels under cardiac stress conditions. Since our previous observation suggested that the post ischemic cardioprotective effect of T3 is associated with the up-regulation of a signature of miRNAs repressed by IR [17], we then tested whether T3-modulated miRNAs might target Brg1. An in silico analysis revealed that among the T3-modulated miRNAs miR-208a-5p, 338-3p, 133a-3p, and 133b exhibited the highest score to target both human and rat Brg1 at multiple positions in the entire mRNA sequence (Figure 2c and Appendix A). In the light of its strategic position within the Myh6 gene, we decided to focus on miR-208a for further in vitro validation of the regulatory circuit. As shown in Figure 2d, miR-208a over-expression resulted in a drastic down-regulation of Brg1 transcript, which was paralleled by a significant 2.2 fold increase of Mhrt expression levels and a 4.8 fold increase of Myh6/Myh7 ratio.

### 3.2. T3 Modulation of Mhrt Promoter and Brg1 Activity Relies on Chromatin Remodeling

To assess the role of T3 in the transcriptional activation of Mhrt, we performed reporter assays in primary rat cardiomyocytes. Since several reports highlight a chromatin remodeling-mediated activity of TH, the full length Mhrt promoter was cloned into two different luciferase reporter plasmids: pREP4 (that is able to undergo chromatinization after transfection into mammalian cells) and pGL3 (that does not allow DNA chromatinization and therefore features a naked promoter). Through this strategy we could test the ability of T3 to distinguish chromatinized from naked Mhrt promoter in cardiomyocytes. The effect of Brg1 over-expression alone or in combination with T3 was estimated by cotransfecting the cells with a Brg1 expression vector. As shown in Figure 3a (left), T3 induced a robust increase of the chromatinized reporter activity while counteracting the inhibitory effect of Brg1. On the contrary, T3 was unable to activate the naked Mhrt promoter and to oppose the Twenty-four hours following ischemia Brg1 inhibitory function (Figure 3a, right).

To further our understanding of the T3 mechanism of action, we then performed an in silico analysis to find putative thyroid hormone receptor (THR) binding sites within the Mhrt promoter. Recently it has been demonstrated that the 3′ terminal portion of the mouse Mhrt promoter, extending from position −441 to +143 (a3-a4-Mhrt), is necessary and sufficient for both Mhrt promoter activation and Brg1-dependent Mhrt repression [3]. To investigate the role of T3 in the regulation of this critical region and to explore potential interplay with Brg1 activity, we decided to restrict the analysis to this DNA sequence. By using the Promo transcription factor search tool, we identified two putative sites with the highest probability to be thyroid hormone response elements (TREs). The first one (site A), located from −265 to −254, is predicted to be a thyroid hormone receptor beta (TRB)-specific binding site (Figure 3b); the second one (site B), located from −389 to −381, is predicted to be a thyroid hormone receptor alpha (TRA)-specific binding site (see Figure 3b). To validate the truthfulness of these TREs, the reporter plasmids containing chromatinized or naked Mhrt promoter were subjected to site-specific mutagenesis and transfected in cardiomyocytes in the presence or absence of T3 treatment. The importance of each site for the function of T3 on Mhrt promoter was assessed by using single or double mutants with nucleotides substitutions at site A (Mut-A), site B (Mut-B), or both (Mut-AB) in order to selectively destroy one or both the putative THR binding sequences (see Figure 3b). The effect of Brg1 over-expression alone or in combination with T3 was also evaluated. The results obtained with the chromatinized and naked plasmids are reported in Figure 3c,d, respectively. In the former condition, the T3 dependent Mhrt activation was halved either by Mut-A and Mut-B and further decreased by the double mutation, even though not completely abolished (Figure 3c). Interestingly, Brg1-dependent Mhrt repression was critically affected by Mut-A, resulting in a threefold increase of promoter activity with respect to the corresponding wild type (Figure 3c). Mut-B did not influence Brg1 function when present alone but seemed to rescue the Brg1 inhibitory effect when present along with Mut-A (Figure 3c). The impact of T3 and Brg1 co-treatment on Mhrt expression was roughly the average of the single treatment in each reporter assay (Figure 3c). A completely different scenario emerged when naked mutant promoters were transfected (Figure 3d). As observed with the WT naked plasmid and irrespectively from the presence of mutations, T3 lost the ability to modulate Mhrt activity that rather resulted slightly reduced in comparison to the unstimulated control. On the contrary, neither T3 nor Mut-A, B, or AB were able to substantially affect the inhibitory action of Brg1 (Figure 3d).

## 4. Discussion

A reduction of the Myh6/Myh7 expression ratio is a well-established molecular marker of maladaptive cardiac remodeling and a target of therapeutic intervention to contrast cardiovascular disease evolution [1,2,3,4]. TH have been previously shown to exert antihypertrofic and anti-remodeling effects via modulation of microRNAs and epigenetic signaling [17,18,21,22]. In this study, thanks to the novel insights on the biology of the LncRNA Mhrt, we have further extended our knowledge on the cardioprotective pathways underlying the T3-dependent orchestration of MHC isoforms switch, including the cross-talk with Brg1 and chromatin remodeling, under physiological and pathological conditions.

Here we have identified the rat Mhrt coding sequence and described the critical involvement of TH in the regulation of its expression. The presence of a TH-sensitive natural antisense transcript (NAT) within the MHC locus has been previously documented in rat myocardium [23,24,25]. However, with respect to the newly identified rat Mhrt sequence, the NAT regulatory molecule spans a different region of the MHC locus and is suggested to act via an alternative mechanism involving RNA-RNA interference.

To the best of our knowledge, this is the first demonstration that a timely reconstitution of euthyroidism by an early and short-term T3 replacement is able to prevent the drop of cardiac Mhrt levels following an acute ischemic insult. Mechanistically, T3 directly activates Mhrt promoter in a chromatinization-dependent manner. THRs are considered master regulators of chromatin structure via recruitment of different chromatin modifying enzymes [22]. Accordingly, our findings are suggestive of a model whereby T3 interacts with chromatin at the Mhrt gene locus and activates transcription via regulation of chromatin modification. In particular, the results obtained after site-directed mutagenesis highlighted two still unknown putative TREs (indicated with A and B) located in the a3-a4 region of the Mhrt promoter. Both site A and B are of utmost importance to confer responsiveness to T3 and for the T3-dependent inhibition of the Brg1function. However, we could still measure a modest but significant T3-stimulated residual promoter activity, even in the AB double mutant, suggesting that other sites, within or outside the a3-a4 DNA region, may be involved in such a complex regulation. In this regard, a pioneering study reported the presence of a TH-sensitive retinoic acid receptor binding site in a distal domain apart from the a3-a4 regulatory region [24], which might explain our data.

Interestingly, T3 replacement under cardiac stress condition is also associated with the reversal of the Brg1 expression pattern. This previously unappreciated action of T3 is probably an indirect consequence of a more general systemic effect activated by cardiac stress. To better clarify the matter, an in silico analysis retrieved Brg1 as a putative target of a signature of T3-regulated miRNAs that are differentially expressed in the AAR of the IR myocardium and confer protection against noxious pathways of acute stress [17]. In particular, miR-208a-5p, 338-3p, 133a-3p, and 133b, that are up-regulated by T3 replacement in the post IR setting, are predicted to also interact with both human and rat Brg1 transcripts, indicating the translational potential of the finding. In accordance with the in silico prediction, the over-expression of miR-208a led to the antithetical down-regulation of Brg1 levels from the one side and up-regulation of Mhrt and Myh6/Myh7 ratio from the other side. Such a first validation of the miR-208a/Brg1 regulatory circuit uncovers a further paradigm for the T3-dependent epigenetic modulation of myosin expression, which may be of clinical relevance.

In addition to the indirect regulation of gene expression, part of the T3 inhibitory effects on Brg1 function seems to occur through some kind of interference between THR and Brg1 chromatin remodeling complex, especially at site A. Although chip analyses are mandatory to confirm this hypothesis, our interpretation is corroborated by previous findings: (1) Brg1 is recruited at nuclear hormone response elements and can directly interact with THR to regulate gene expression either during mammalian development and in the adult life [26,27,28]; (2) epigenetic marks of active chromatin are enriched following T3 treatment in a region embracing site A [29]; (3) in the same region repressive chromatin remodeling marks are enriched by Brg1while epigenetic marks of active chromatin are down-regulated [3].

An unexpected finding of the present work is the ability of Brg1 to exert its repressive action on Mhrt promoter even within the naked plasmid. Brg1 has always been characterized as a chromatin-remodeling factor and its ability to bind only chromatinized Myh6 promoter has been also shown [3]. These apparently contradictory results might be reconciled by a recent work unraveling a previously unrecognized capacity of the Brg1 bromodomain (BRD) to interact with naked DNA, which appears to be a gain of function in the BRD of Switch/Sucrose-Non-Fermentable (SWI/SNF) ATPases of higher eukaryotes [30]. It has been hypothesized that such activity might contribute to chromatin targeting or positioning of the BAF complex on the nucleosomes [30]. However, further studies are needed to clarify the in vivo physiological relevance of this function of Brg1, its involvement in MHC switch, and its relation with the effect of T3.

## 5. Conclusions

In conclusion, the T3-dependent regulation of the Mhrt/Brg1 axis appears to be a novel cardioprotective mechanism that contributes to explain why TH treatment is associated with the reversal of pathological cardiac hypertrophy in favor of more physiological cardiac growth [10]. According to our results, we propose a model whereby T3 affects a network of cross-talking epigenetic modifiers that act at the MHC locus and regulate each other to maintain the proper MHC fiber composition of the adult heart. In this scenario, chromatin remodeling systems and non-coding RNAs seem to play a major role (Figure 4). The overall data may be helpful in translating T3 replacement into clinical practice. Indeed, Mhrt repression and Brg1 up-regulation have been documented in patients with hypertrophic, ischemic, or idiopathic cardiomyopathy, thus suggesting a conserved mechanism of human cardiomyopathy [3,4]. Both in experimental setting and in the clinical arena, pathologically remodeled hearts due to pressure overload or myocardial infarction exhibit a local or systemic low T3 state (LowT3S) with alterations in gene expression program that are similar to the genetic profile of fetal or hypothyroid hearts [12,31,32]. Such reduced T3-signaling in the stressed heart may aggravate adverse remodeling by releasing the T3-dependent inhibition on Brg1, thus fostering the maladaptive MHC switch (Figure 4). It is conceivable that a T3 replacement strategy in cardiac patients with a LowT3S may blunt the progression towards heart failure by restoring the physiological Mhrt/Brg1 cardiac balance.

## Figures and Tables

**Figure 1 cells-09-02155-f001:**
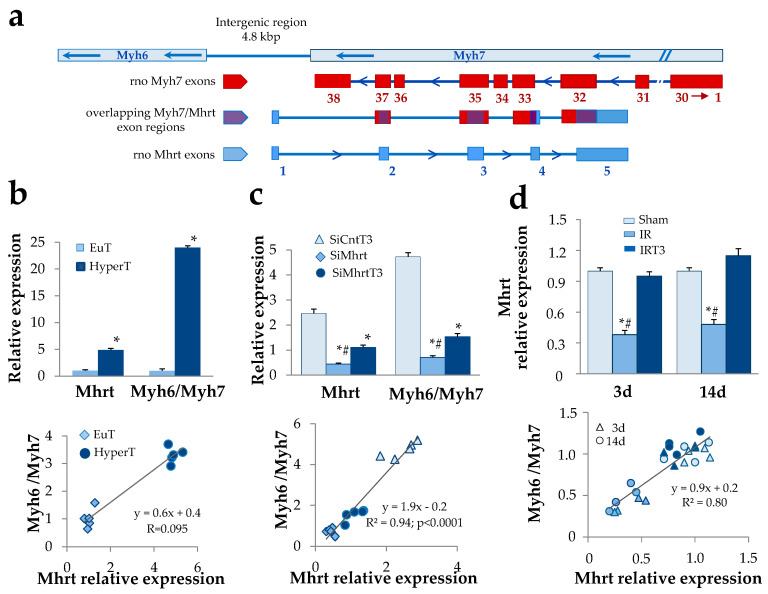
Thyroid hormone-dependent regulation of Mhrt greatly affects Myh6/Myh7 expression ratio. (**a**) Schematic illustration of the rat Mhrt, Myh6, and Myh7 gene organization. Red boxes represent Myh7 exons; blue boxes represent Mhrt exons; the overlapping Myh/Mhrt exon regions are depicted in violet. (**b**) Quantification of cardiac Mhrt expression and correlation with Myh6/Myh7 expression ratio in euthyroid rats (EuT) and in hyperthyroid rats (HyperT). (*n* = 5; * *p* < 0.0001 HyperT vs. EuT). (**c**) Quantification of Mhrt expression and correlation with Myh6/Myh7 expression ratio in cultured rat neonatal cardiomyocytes (NRCM). Twenty-four hours after transfection with SiRNA specific for Mhrt (SiMhrt) or control SiRNA (SiCnt), the NRCM were maintained for 48 h in the presence of T3 at physiological concentration (3 nM) or T3 vehicle. Light blue, triangle = SiCntT3; blue, rhombus = SiMhrt; dark blue, circle = SiMhrtT3 (*n* = 5 different cell batches; * *p* < 0.0001 vs. SiCnt, ^#^
*p* ≤ 0.01 vs. SiMhrtT3). (**d**) Quantification of Mhrt expression and correlation with Myh6/Myh7 expression ratio in an in vivo rat model of cardiac ischemia/reperfusion (IR). Twenty-four hours after IR, rats were subjected to 48 h infusion of T3 at replacement dose or to infusion of T3 vehicle for the same time. Mhrt and Myh6/Myh7 levels in the area at risk (AAR) of the left ventricle were quantified at 3 d and 14 d post IR. Color code: light blue = Sham; blue = IR; dark blue = IRT3; shape code: triangle = 3 d post IR; circle = 14 d post IR (*n* = 4 biological repeats for each experimental condition; * *p* < 0.0001 vs. Sham, ^#^
*p* < 0.0001 vs. IRT3).

**Figure 2 cells-09-02155-f002:**
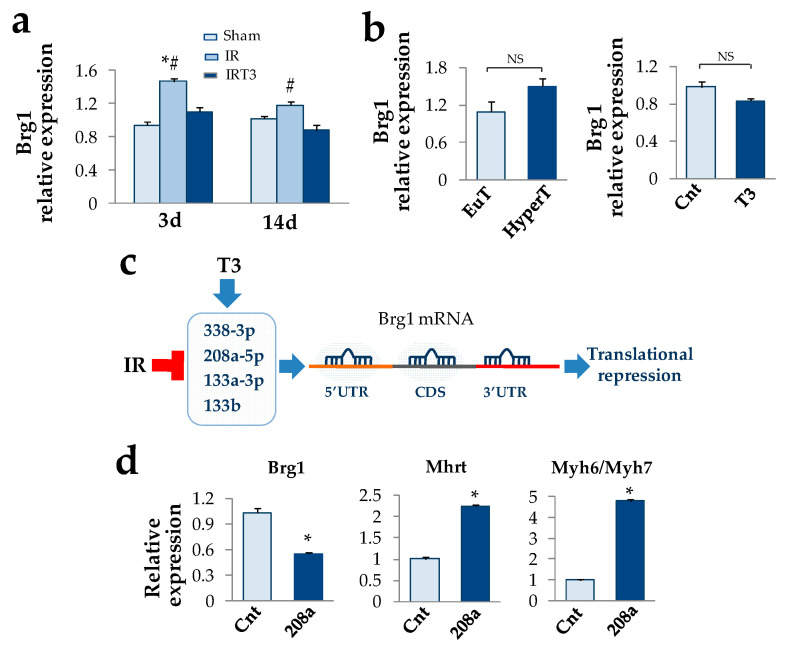
T3 replacement blunts Brg1 over-expression indirectly under acute cardiac IR stress. (**a**) Quantification of Brg1 expression in the AAR of the rat left ventricles at 3 d and 14 post IR in the presence and absence of early (24 h post IR) and short term (48 h) T3 replacement. (*n* = 4 biological repeats for each experimental condition; * *p* < 0.0001 vs. Sham, ^#^
*p* ≤0.01 vs. IRT3). (**b**) Neither experimental hyperthyroidism nor T3 administration to neonatal rat cardiomyocyte cultures are able to modulate Brg1 expression. (**c**) T3-differentially expressed (DE) miRNAs that are predicted to target both human and rat Brg1 mRNA at 5′ untranslated region (5′UTR), coding sequence (CDS), or 3′ untranslated region. (**d**) In accordance with the in silico prediction, over-expression of miR-208a mimic results in Brg1 down-regulation and increase of Mhrt expression and Myh6/Myh7 ratio (*n* = 4 biological repeats for each experimental condition; * *p* < 0.0001 vs. control mimic).

**Figure 3 cells-09-02155-f003:**
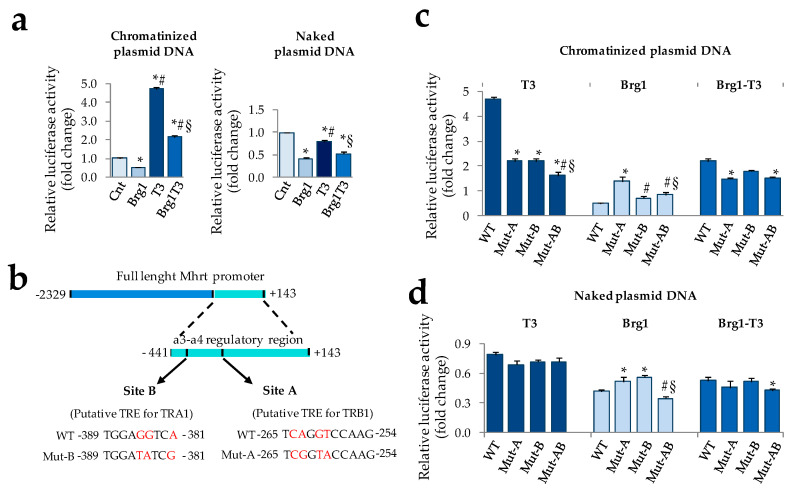
Modulation of Mhrt promoter activity by T3 requires DNA chromatinization and two putative thyroid hormone responsive elements (TREs) (**a**) Dual luciferase assay to evaluate the impact of T3 and Brg1 alone or in combination on the activity of the chromatinized (left) and naked (right) Mhrt promoter in neonatal rat cardiomyocytes. T3 was administered 24 h after transfection at physiological concentration (3 nM) for 48 h (*n* = 5 biological repeats; * *p* ≤0.001 vs. Cnt, ^#^
*p* <0.0001 vs. Brg1, ^§^
*p* <0.0001 vs. T3). (**b**) Schematic illustration of the mouse a3-a4 regulatory region of Mhrt promoter and the putative TREs for TRA1 and TRB1. Wild type and mutated sequences are also reported. (**c**) Dual luciferase assay to evaluate the impact of T3 and Brg1 alone or in combination on the activity of wild type and mutated chromatinized Mhrt promoter in cardiomyocytes (*n* = 4 biological replicates. * *p* ≤ 0.001 vs. wild type (WT), ^#^
*p* ≤ 0.002 vs. Mut-A, ^§^
*p* ≤ 0.003 vs. Mut-B). (**d**) Luciferase assay to evaluate the impact of T3 and Brg1 alone or in combination on the activity of wild type and mutated naked Mhrt promoter in cardiomyocytes (*n* = 4 biological replicates. * *p* ≤ 0.04 vs. WT, ^#^
*p* ≤ 0.001 vs. Mut-A, ^§^
*p* = 0.01 vs. Mut-B).

**Figure 4 cells-09-02155-f004:**
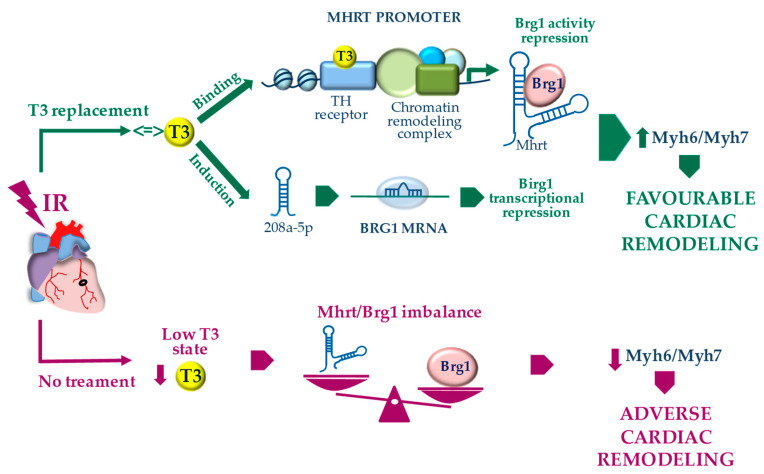
Proposed model for the T3-dependent regulation of Myh7 to Myh6 switch through the modulation of the Mhrt/Brg1 axis following acute myocardial IR.

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
