# Peer review of "T3 Critically Affects the Mhrt/Brg1 Axis to Regulate the Cardiac MHC Switch: Role of an Epigenetic Cross-Talk"

_cells, 2020, doi:10.3390/cells9102155_

Round 1

Reviewer 1 Report

The manuscript by Forini et al. used in silico analyses and site-directed mutagenesis combinated with gene reporter assays and loss or gain of function assays in cultured cardiomyocytes and evidenced a new effect of T3 in the Mhrt/Brg1 axis, suggesting that this regulates the cardiac MHC shift observed in some conditions. The approach of this manuscript is very interesting. Overall the authors are to be complimented, the manuscript has merit because it is a well-conducted study and its scientific soundness is reasonable. This group has extensive expertise in studying the thyroid hormone mechanisms associated to cardiac hypertrophy. However, there are a few aspects of the manuscript that detract from the merit of the study. Some details and further explanations need to be addressed.

-The Introduction might be more contextualized.

Methodology:

- There is no mention to injection of T3 instead of thyroxine in the Wistar rats.

- The authors cite that "10 uM of Brdu was added to the culture medium" to prevent the proliferation of fibroblasts. However, how can they ensure that this drug, which is inserted in the DNA strand, does not interfere with the results?

- Authors cite that "FBS was deprived of TH with standard charcoal stripping procedures". This method needs to described. How to ensure selectivity for only TH to be stripped; do you have these controls about the presence of growth factors in the medium like found in the control condition??

Discussion:

- Although in silico analysis has exhibited high score to T3-modulated miRNAs (miR 208 and miR 133), previous studies have described the involvement of miR208, miR 133, miR-1 in the cardiomyocyte hypertrophy induced by T3 (Diniz GP, 2017; 2015; 2013). These studies should be considered to shed new light in the discussion section.

- The first statement of Discussion, relative to Myh6/Myh7 expression ratio should be referenced.

- The authors mention that “T3 replacement under cardiac stress condition is associated to the reversal of the Brg1 expression pattern.” This statement can be better discussed in the light of the knowledge related to adrenergic nervous system or renin-angiotensin system activation by T3 in the heart.

Reviewer 2 Report

The manuscript ‘T3 critically affects the Mhrt/Brg1 axis to regulate the cardiac MHC switch: role of an epigenetic cross-talk’ aims to evaluate the role of T3 on the Mhrt/Brg1 axis. Overall, the manuscript presents data that are important to the field, yet a few points need to be clarified. The specific points that resulted in this conclusion are listed below. 

Lines 90-92: Keep font size consistent.

Line 99: Change ‘prior NRC’ to ‘prior to NRCM’.

Line 111: Does Myh7 need to be changed to Mhrt? How does a siRNA for Mhrt silence Myh7?

While the number of cell culture plates used is stated in the figure legends, this also needs to be included in the material and methods section.

Throughout the manuscript, the phenotype of the cells is never discussed. Some of these cells were part of an experiment with time points of 3 and 14 days. Do the cells change in phenotype? Does the transfected construct alter proliferation, confluency or differentiation of the cells?

Figure 1. Some of the shapes in the graphs are not defined. For example, what does a dark blue triangle represent?

Line 327: Change ‘activate Mhrt promoter’ to ‘activates the Mhrt promoter’.

Line 348: Change ‘lead’ to ‘led’.

Appreciate that the authors are concise with their results and discussion and do not make more of the data than what they are. Figure 4 is helpful for bringing everything together and proposing a role for T3.

Round 2

Reviewer 1 Report

The authors have answered all questions addressed previously.